# Roles of Thyroid Hormone-Associated microRNAs Affecting Oxidative Stress in Human Hepatocellular Carcinoma

**DOI:** 10.3390/ijms20205220

**Published:** 2019-10-21

**Authors:** Po-Shuan Huang, Chia-Siu Wang, Chau-Ting Yeh, Kwang-Huei Lin

**Affiliations:** 1Department of Biochemistry, College of Medicine, Chang-Gung University, Taoyuan 33302, Taiwan; leo_6813@msn.com; 2Department of Biomedical Sciences, College of Medicine, Chang-Gung University, Taoyuan 33302, Taiwan; 3Department of General Surgery, Chang Gung Memorial Hospital, Chiayi 61363, Taiwan; wangcs@cgmh.org.tw; 4Liver Research Center, Chang Gung Memorial Hospital, Linkou, Taoyuan 33302, Taiwan; chauting@adm.cgmh.org.tw; 5Research Center for Chinese Herbal Medicine, College of Human Ecology, Chang Gung University of Science and Technology, Taoyuan 33302, Taiwan

**Keywords:** oxidative stress, microRNA, thyroid hormone, liver cancer

## Abstract

Oxidative stress occurs as a result of imbalance between the generation of reactive oxygen species (ROS) and antioxidant genes in cells, causing damage to lipids, proteins, and DNA. Accumulating damage of cellular components can trigger various diseases, including metabolic syndrome and cancer. Over the past few years, the physiological significance of microRNAs (miRNA) in cancer has been a focus of comprehensive research. In view of the extensive level of miRNA interference in biological processes, the roles of miRNAs in oxidative stress and their relevance in physiological processes have recently become a subject of interest. In-depth research is underway to specifically address the direct or indirect relationships of oxidative stress-induced miRNAs in liver cancer and the potential involvement of the thyroid hormone in these processes. While studies on thyroid hormone in liver cancer are abundantly documented, no conclusive information on the potential relationships among thyroid hormone, specific miRNAs, and oxidative stress in liver cancer is available. In this review, we discuss the effects of thyroid hormone on oxidative stress-related miRNAs that potentially have a positive or negative impact on liver cancer. Additionally, supporting evidence from clinical and animal experiments is provided.

## 1. Introduction

Hepatocellular carcinoma (HCC) is an inflammation-related cancer, with the majority of cases occurring in the context of hepatic injury and inflammation [1]. The risk factors correlated with HCC include chronic inflammation due to viral infection (such as hepatitis B virus (HBV) and hepatitis C virus (HCV)), excessive intake of alcohol, metabolic disease, non-alcoholic steatohepatitis (NASH), bacterial infection, type 2 diabetes, smoking, and chemical exposure [2]. Both HCC and the associated risk factors are significantly correlated with oxidative stress. Oxidative stress occurs when excessive production of reactive oxygen species (ROS) overpowers intrinsic antioxidant defense mechanisms. Accumulating levels of ROS can cause extensive damage to biological molecules, leading to cell injury, loss of function, development of cancer, and even death [3]. Therefore, elucidation of the relationship between oxidative stress and cancer is of clinical importance. Among the potential mechanisms involved in carcinogenesis, the pathways triggered by oxidative stress-induced microRNAs (miRNA) have been widely investigated. MiRNAs are small endogenous non-coding RNA molecules that regulate multiple gene expression at the post-transcriptional level. These molecules suppress messenger RNA through binding to stretches of complementary sequences [4,5]. The potential associations of miRNAs with human disease are widely documented. As crucial regulators of gene expression, miRNAs thus present promising candidates for biomarkers and treatment strategies.

Thyroid hormones play major roles in cell growth, development, and metabolism. Considerable research supports a relationship between the thyroid hormone and pathophysiology of various cancer types. Thyroid hormones exert their effects on cancer cells through either genomic or non-genomic pathways and their dysregulation has significant effects on cancer development and progression. Hypothyroidism is reported to contribute to liver carcinogenesis [6]. Notably, both hyperthyroidism and hypothyroidism appear to be associated with oxidative stress in animal and human diseases, indicating involvement of the thyroid hormone in disease progression [7]. Preliminary data from recent studies focusing on the potential relationship between miRNAs associated with oxidative stress and dysregulation of thyroid hormone in liver cancer progression are comprehensively summarized in the current review.

## 2. Effect of Thyroid Hormone on the Role of Oxidative Stress-Related microRNAs in Liver

### 2.1. Oxidative Stress Promotes HCC Progression

Liver cancer is the second leading cause of cancer-related deaths worldwide. Hepatocellular carcinoma (HCC), a type of inflammation-related cancer with >90% cases associated with hepatic injury and inflammation, is the most common primary malignant tumor type [8]. The incidence of HCC is highly correlated with inflammatory risk factors, such as hepatitis B virus (HBV), hepatitis C virus (HCV), liver disease (non-alcoholic fatty liver disease/non-alcoholic steatohepatitis), habitual drinking (high alcohol exposure), obesity, type 2 diabetes (T2D), and aflatoxin exposure [9,10,11,12].

Oxidative stress additionally plays an important role in HCC development. Excess ROS levels induce liver DNA injury, in turn leading to increased fatty liver, hepatitis B/C, liver cirrhosis, and consequently, HCC [13]. Oxidative stress is defined as an imbalance between production of reactive oxygen species (ROS) and antioxidant capacity of the cell, which causes damage to biomolecules, such as DNA, lipids, and proteins [14].

ROS simultaneously affect a series of signaling cascades and mediate the regulation of several transcription factors that control the expression of various genes involved in cell survival, proliferation, invasion, and metastasis [15,16,17,18,19]. Common ROS species include superoxide anion (O^2−^), hydrogen peroxide (H_2_O_2_), hydroxyl radical (OH^−^), singlet oxygen (1 O_2_), and ozone (O_3_) [20,21]. Reactive species induce nicks in DNA and failure in mechanisms to repair DNA damage that lead to HCC.

ROS can react with cellular biomolecules, yielding oxidatively modified DNA products that eventually induce cell damage and death. For instance, protein carbonyl and 8-hydroxydeoxyguanosine (8-OHdG), the well-known oxidatively modified molecular products of proteins and DNA, are associated with poor survival in HCC patients [16].

The inflammation risk factors, HCV and HBV infection, cause malignant degeneration by induction of oxidative stress that is critical in HCC. Oxidative stress is present to a greater degree in HCV infection than other inflammatory liver diseases and proposed as a major mechanism of liver injury in patients with chronic hepatitis C [22]. The core protein of HCV, which induces excess ROS production through adjustment of mitochondrial electron transport and mitochondria, is a primary target of ROS. Therefore, damage to mitochondria via ROS induced by HCV presents a potential mechanism underlying the development of HCC [23].

In addition to HCV, HBV infection markedly increases the risk of development of HCC. Among the viral proteins, HBV encoding HBV X protein (HBx) appears to have the greatest oncogenic potential in HCC. Similar to HCV core protein, HBx is associated with mitochondria, leading to augmented ROS production and induction of oxidative stress in hepatocytes [24]. The key mechanisms used by HBx, such as inhibition of high-mobility group protein box1 (HMGB1) expression and generation of ROS via the NF-κB signaling pathway, are discussed in an earlier report [25].

Non-alcoholic fatty liver disease (NAFLD) is a complex disorder characterized by excessive lipid accumulation the in liver, controlled by multiple metabolic factors, that is often diagnosed in conjunction with obesity, type 2 diabetes (T2D), and hyperlipemia [26]. Among the numerous mechanisms underlying NAFLD pathogenesis, redox imbalance is suggested to be the most significantly correlated factor to HCC progression. In addition, conditions such as metabolic oxidative stress, cell autophagy, and inflammation induce more severe nonalcoholic steatohepatitis (NASH) progression [26]. In patients with NASH, the activities of mitochondrial respiratory chain complexes are decreased in liver tissue, resulting in reduced glutathione expression and consequent activation of the c-Jun N-terminal kinase (JNK)/c-Jun signaling pathway by oxidative stress that induces cell death in steatotic liver [27].

The issue of whether risk factors directly induce or are subject to oxidative stress to increase their effects remains to be established. However, the findings to date suggest that oxidative stress exerts harmful effects on liver cells through inducing lesions. Elucidation of the underlying mechanisms should facilitate the development of effective strategies to manage HCC.

### 2.2. Roles of microRNAs Correlated with Oxidative Stress in HCC

MicroRNAs (miRNAs) are small non-coding RNA molecules that regulate >70% human genes at the post-transcriptional level. The average miRNA length is ~21–23 nucleotides. DNA sequences are transcribed into primary miRNAs (pri-miRNA) and processed into precursor miRNAs (pre-miRNA) in the nucleus and mature miRNAs in the cytoplasm. In most cases, miRNAs interact with a specific sequence at the 3′ untranslated region (UTR) of target mRNAs to induce translational repression via post-transcriptional regulation of cleavage or simply suppressing translation [28,29].

Accumulating studies support the importance of a series of oxidative stress-induced miRNAs in progression of carcinogenesis (Table 1). For instance, using the robust rank aggregation (RRA) method, miR-34a-5p, miR-1915-3p, miR-638, and miR-150-3p were shown to be upregulated under conditions of H_2_O_2_ treatment as oxidative stress-responsive miRNAs in HCC cell lines [30]. The functions of these four miRNAs were further predicted using the TargetScan web tool and Gene Ontology (GO) pathway enrichment analysis. All four miRNAs were closely related to anti-apoptosis pathways and p53 signaling, clearly demonstrating a significant association between the p53 pathway and oxidative stress [31].

The importance of miRNAs in progression of chronic liver diseases to HCC is recognized. MiRNAs act as key mediators in the development of a number of cancer types owing to their involvement in inflammation and oncogenesis processes. Several miRNAs showing altered expression patterns in HCC and oxidative damage have been identified, including miRNA-92, miRNA-145, miRNA-199a, miRNA-199b, miRNA-195, and miRNA-122a [27].

In recent years, several miRNAs have been extensively investigated and their functions in association with oxidative stress determined. MiR-26a is reported to play a dual role in HCC. Considerable research has confirmed its activity as a tumor suppressor in HCC that inhibits proliferation, migration, and invasion by targeting F-box protein 11 (FBXO11), an E3 ubiquitin ligase, and type II methyltransferase [47,48]. DNA methyltransferase 3b (DNMT3B) is another direct target of miR-26a. Inhibition of DNMT3B associated with miR-26a upregulation led to a similar tumor suppressor effect in HCC cells [49]. In contrast, other studies suggest that miR-26a has potential oncogenic function in HCC. For instance, therapeutic miR-26a delivery suppresses tumorigenesis in an animal liver cancer model while other studies demonstrated that miR-26a expression promotes HCC cell migration and invasion in vivo. Another earlier in-depth study reported that miR-26a promotes cell migration and invasion by inhibiting the phosphatase prime time entertainment network (PTEN) [50,51]. Based on the metabolic perspective, increasing free fatty acid (FFA) supply into liver cells caused oxidative stress by ROS and lipid peroxidation generated during the metabolism of these accumulating fatty acids [52]. Recently, regulatory and protective roles of miR-26a on lipid metabolism and progression of NAFLD in human HepG2 cells loaded with FFA have been demonstrated. Upregulation of miR-26a resulted in the downregulation of triglyceride (TG), total cholesterol (TCL), and malondialdehyde (MDA) through modulation of mRNA levels of genes involved in lipid homeostasis, ER stress, inflammation, and fibrogenesis [36]. Additionally, miR-26a targets different metabolic relative genes involved in fatty acid and cholesterol metabolism and insulin signaling, such as ACSL3, ACSL4, PKCδ, PKCθ, GSK3β, and SERBF1, suggesting a crucial role in preventing development of metabolic disease [53]. Notably, these liver-related lipid metabolism abnormalities are strongly associated with oxidative stress in liver cells [54,55].

MiR-155 acts as a multifunctional oncogenic miRNA in different human cancer types, including breast, pancreatic, and liver cancer [56,57,58]. The miRNA promotes proliferation, invasion, and migration in HCC by directly targeting and inhibiting PTEN. The negative correlation between miR-155 and PTEN is significantly associated with TNM stage in HCC [56]. MiR-155 additionally inhibits Forkhead box O3 (FoxO3a) expression to suppress downstream apoptotic gene B-cell lymphoma-2 (Bcl-2)-interacting mediator of cell death (BIM) and suppresses cleavage of caspase-3 and caspase-9, consequently inhibiting HCC cell apoptosis and facilitating proliferation [59]. Furthermore, high expression of miR-155 is associated with poor survival, and in combination with Alpha-fetoprotein (AFP) shows higher sensitivity and specificity as a biomarker panel for diagnosis of HCC, compared with a single marker [60]. However, conflicting results on the role of miR-155 in lipid metabolism have been reported to date. Suppression of miR-155 in peripheral blood may be utilized as a novel biomarker for NAFLD screening. The transcription factor, Liver X Receptor α (LXRα), that interacts with the promoter region of sterol regulatory element-binding protein (SREBP)-1c, has been identified as a direct target of miR-155 [37,61]. Other studies have highlighted a reduction in alcohol-induced fat accumulation in miR-155 knockout mice, associated with increased Peroxisome proliferator-activated receptor response element (PPRE) binding to the miR-155 target gene, Peroxisome proliferator-activated receptor (PPAR)α [62]. However, further studies are required to confirm the finding that miR-155 participates in lipid accumulation in liver, inducing generation of oxidative stress.

### 2.3. Role of Thyroid Hormone and Its Receptor in HCC

Thyroid hormone, 3,3’,5-tri-iodo-l-thyronine (T3), is a key mediator of multiple physiological processes, including cell development, differentiation, metabolism, and growth [38]. The pituitary gland secretes thyrotropin, which influences the thyroid gland to synthesize thyroid hormone mainly precursor T4. T4 moves across the cell membrane of responsive cells by specific transporters, including the monocarboxylate anion transporters 8 and 10 (MCT8 and MCT10), and is converted to the active T3 by type I 5’-deiodinase (DIO) 1 and 2, leading to increased levels of T3 [63]. T3 controls metabolic activities related to anabolism or catabolism, including carbohydrates, proteins, lipids, and damaged organelles in cells to maintain homeostasis under different physiological conditions [24]. To implement genomic effects, cytoplasmic T3 translocates to the nucleus and binds to specific high-affinity thyroid hormone receptors (TR) associated with thyroid hormone response elements (TRE) on DNA, thereby affecting transcriptional levels of downstream genes [14]. Typical TREs within promoter regions of downstream genes contain two half-site sequences (A/G)GGT(C/A/G)A in palindromic (Pal), direct repeat (DR), or inverted repeat arrangements (IP) recognized by TR. TRs bind to their respective TREs as monomers, homodimers, or heterodimers with retinoid X receptors (RXR). TRs usually form heterodimers with the RXR to interact with TREs within the promoter regions of target genes. Human TRs are encoded by two distinct genes, THRA (TRα) and THRB (TRβ), located on human chromosomes 17 and 3 [64]. Different TRs are composed of similar domains, including amino-terminal A/B domain to recruit regulatory proteins; central DNA-binding domain (DBD), or C region, which displays high affinity for DNA sequences of TREs; linker D region, which is necessary for nuclear translocation of the receptor; and carboxy-terminal ligand-binding domain (LBD), which interact with thyroid hormones [63,65]. In humans, TRβ/T3 regulates the metabolic activity of body and it is the major receptor isoform expressed in liver; in contrast, TRα is expressed mainly in the heart, skeletal muscle, adipose tissues, and specifically mediates adaptive thermogenesis [66].

Owing to its critical regulatory function in cellular homeostasis, imbalance of thyroid hormone in the body is highly associated with multiple chronic diseases including obesity, diabetes, cardiovascular, and liver disorders. The liver is the most important thyroid hormone target organ associated with cellular metabolic functions, such as hepatic fatty acid and cholesterol synthesis and metabolism. Hypothyroidism has been associated with increased serum expression of triglycerides and cholesterol as well as hypercholesterolemia or non-alcoholic fatty liver disease (NAFLD) [24,67]. Prevention of cardiovascular disease occurrence is important in patients with low-serum high-density lipoprotein cholesterol (HDL-C) due to thyroid dysfunction [68].

In addition to its effects on metabolism, thyroid hormone suppresses HCC development by protecting hepatocytes from HBx-induced damage through regulating mitochondrial quality control to suppress HBx protein stability. Mitochondrial quality maintenance by T3 prevents HBx-induced hepatocarcinogenesis and attenuates HCC progression [25,69]. In an earlier study, liver disease patients diagnosed with hepatic cirrhosis triggered by hepatitis B or C were screened for thyroid function status. The T3 levels of patients were lower than the normal range, suggesting that the serum T3 concentration is a good index of hepatic function, decreasing the severity of liver damage [70].

Several studies have demonstrated that treatment with T3 analogs can prevent hepatic steatosis and hepatitis. The thyroid hormone has potential therapeutic applications in hepatitis B and C, and T3 analogs may be effectively used as an alternative strategy to prevent HCC [71].

### 2.4. Thyroid Hormone Induces an Anti-Oxidative Stress Effect in Hepatocytes Mediated by microRNAs

Hypermetabolic effects of thyroid hormones as the major endocrine regulators of metabolic rate are well documented. Thyroid hormones have a profound impact on mitochondria, the organelles predominantly responsible for cellular energy metabolism, and are correlated with O_2_ consumption and consequent ROS generation [72]. Effects of thyroid hormone on redox signaling to protect cellular function are documented. The pathways affected by thyroid hormone generally fall into two broad categories: Genomic and non-genomic. ROS production leads to activation of the redox-sensitive transcription factors nuclear factor-κB (NF-kB), signal transducer and activator of transcription 3 (STAT3), signal transducer and activator of transcription 1 (STAT1), and nuclear factor erythroid 2-related factor 2 (Nrf2), promoting cell protection and survival mechanisms. Functions of the thyroid hormone include enhancement of homeostatic potential, through induction of antioxidant, anti-apoptotic, and anti-inflammatory gene expression, and higher detoxification capabilities and energy supply through AMP-activated protein kinase (AMPK) upregulation [73]. Thyroid hormone additionally regulates miRNAs that promote antioxidant capacity in the liver to prevent HCC progression.

In a previous study, our group used qRT-PCR array to explore the expression patterns of different miRNAs regulated by thyroid hormone (Table 2 and Table 3) [38,74,75,76]. The functions of potentially important thyroid hormone-regulated miRNAs in HCC and their correlation with oxidative stress are further discussed below.

MiR-214 is dysregulated in many human cancer types including cervical, prostate, and ovarian cancer [91,92,93]. In HCC, miR-214 acts as a tumor suppressor and is used as a potential prognostic marker for overall survival [94,95]. Earlier studies indicate that miR-214 plays a tumor suppressor role by inhibiting proliferation and migration of HCC cells through targeting pyruvate dehydrogenase kinase 2 (PDK2) and plant homeodomain finger protein 6 (PHF6) [80]. Forkhead box protein M1 (FoxM1) is an important transcription factor in the progression of HCC. Direct targeting and downregulation of FoxM1 mRNA by miR-214 inhibits proliferation, migration, and invasion of HCC [81]. In the clinic, miR-214 downregulation is positively associated with higher tumor recurrence and poorer clinical outcomes. Ectopically expressed miR-214 inhibits xenograft tumor growth and microvascularity of tumors and their surrounding tissues via targeting and suppressing its downstream target gene, hepatoma-derived growth factor (HDGF) [94].

Several oncogenic long non-coding RNAs (lncRNA) are correlated with miR-214. Among these, myocardial infarction-associated transcript (MIAT) regulates proliferation and invasion of HCC cells via sponging miR-214 [96]. Plasmacytoma variant translocation 1 (PVT1) lncRNA is increased in HCC tissues and associated with tumor size, histological differentiation grade, and advanced TNM stage. PVT1 has been shown to promote proliferation and invasion of HCC via inhibition of miR-214 expression by interacting with enhancer of zeste homolog 2 (EZH2) [97].

MiR-214 is upregulated by the thyroid hormone through direct interactions with its receptor in the promoter region, leading to repression of the target oncogene, PIM-1, and in turn, suppression of HCC cell proliferation and inhibition of tumor formation [38]. Diethylnitrosamine (DEN) is a typical chemical carcinogen with the potential to cause tumors in multiple organs, such as liver, skin, gastrointestinal tract, and the respiratory system. This significant environmental carcinogen triggers ROS production, resulting in oxidative stress and cellular injury. DEN is considered a complete hepatocarcinogen [98,99,100]. As highlighted previously, thyroid hormone promotes selective autophagy via induction of the death-associated protein kinase 2-Sequestosome 1 (DAPK2-SQSTM1) pathway, thus protecting against DEN-induced carcinogenesis in hepatocytes [101]. Notably, thyroid hormone additionally plays a protective role against DEN-induced HCC through upregulation of miR-214 [38].

Thyroid hormone is a human hormone that mediates the cell differentiation and metabolism and acts as an anti-apoptosis factor upon challenge of thyroid hormone receptor expression in HCC cells with cancer therapy drugs, such as cisplatin, doxorubicin, and tumor necrosis factor-related apoptosis-inducing ligand (TRAIL). Doxorubicin (Dox), a DNA topoisomerase II inhibitor, belongs to the anthracycline anticancer drug family [102]. Dox is widely used to treat lymphoma breast, head-and-neck, prostate, and liver cancers [103,104,105,106]. Dox induces pathogenic mechanisms including apoptosis, oxidative stress, and inflammation, through formation of ROS, reduces anti-oxidative defense, and stabilizes mitochondrial damage [107,108,109]. Thyroid hormone and its receptor signaling pathway promote chemotherapeutic resistance through negatively regulating the pro-apoptotic protein, BCL2-like 11 (BCL2L11/Bim), resulting in Dox-induced metastasis of chemotherapy-resistant HCC cells [110].

In addition, HCV infection promotes mitochondrion-mediated apoptosis through stimulating the upstream ROS/JNK signaling pathway to affect Bax-triggered mechanisms. In brief, HCV-induced ROS/JNK signaling transcriptionally activates Bim expression, which leads to Bax activation and apoptosis induction [111]. Bim is a direct target gene of miR-214 in nasopharyngeal carcinoma (NPC) and other tissues [112,113,114]. One possibility is that the thyroid hormone induces miR-214 to suppress Bim expression through negatively regulating the transcription factor Forkhead box protein O1 (FoxO1) to avoid liver cell apoptosis and ROS-induced stress [110]. In addition to miR-214, there are many miRNAs that have the potential to affect the apoptosis of liver cancer cells, such as miR-155, miR-4417, miR-199a, and miR-122 [59,77,115,116]. Among them, the expression levels of miR-199a and miR-122 are associated with thyroid hormone and oxidative stress (Table 2 and Table 3) [27]. Previous studies have indicated that Dauricine (Dau) is a natural alkaloid, which promoted apoptosis of HCC cells induced by chemotherapeutic reagents. Dau stimulates the expression of miR-199a and results in inhibition of the target gene hexokinase 2 (HK2) and pyruvate kinase M2 (PKM2), resulting in sensitivity to chemotherapeutic reagents, including Cisplatin, Sorafenib, and Isoliensinine in HCC cells [116]. However, thyroid hormones, which are inversely related to miR-199a, may also involve in this action of apoptosis. Interestingly, in addition to the target gene of miR-199a, PKM2, which is strongly associated with apoptosis, is also affected by miR-4417 and miR-122 in liver cancer cells [77,115].

Gemcitabine (GEM) is a commonly used chemotherapeutic agent for HCC that uses oxidative stress induction as a common effector pathway. Overexpression of mitochondrial uncoupling protein 2 (UCP2) causes resistance to GEM. GEM administered alone or in combination with oxaliplatin renders minimal survival benefits to HCC patients. The tumor suppressor activity of miR-214 is activated through targeting UCP2, which may solve the problem of GEM efficacy [117,118]. Combined usage of thyroid hormone combined with GEM could provide new insights into strategies to treat liver cancer based on this novel mechanism of action.

MiR-214 protects red blood cells against oxidative stress by targeting activating transcription factor 4 (ATF4) and enhancer of zeste homolog 2 (EZH2) [39]. Direct targeting of the transcriptional factor, ATF4, by miR-214, attenuates stress responses. Suppression of miR-214 leads to enhanced ATF4 translation and consequently, upregulation of ATF4 protein. Additionally, miRNA-214 is reported to reduce oxidative stress in diabetic nephropathy mediated the ROS/Akt/mTOR signaling pathway [119]. An earlier study by Liu et al. [120] showed that miR-424 inhibits oxidative stress and protects against transient cerebral ischemia injury.

MiR-122 plays a complex role in HBV and HCV infection [34,121,122]. MiR-122 is a liver-specific miRNA that acts as a host factor to increase the abundance of HCV RNA by stabilizing the positive strand of HCV RNA genome and promotes HCV synthesis by binding two sites near the HCV 5’ end and associating with Ago2 [34,123,124,125]. In contrast to its role in HCV infection in HCC, miRNA-122 is significantly downregulated in patients with HBV infection [126,127]. Adenosine deaminases act on RNA-1 (ADAR1), an important gene involved in adenosine to inosine RNA editing and miRNA processing. ADAR1 also plays an anti-viral role against HBV infection by increasing the miRNA-122 level in hepatocytes [128]. In terms of the role of miR-122 in carcinogenesis, this liver-specific miRNA is reported to be dramatically downregulated in most HCCs. The tumor suppressor role of miR-122 in HCC is exerted by targeting the genes involved in cell proliferation, differentiation, apoptosis, and angiogenesis, and its expression is inversely associated with poor prognosis and metastasis [129]. Many studies have demonstrated that miR-122 acts as an important tumor suppressor through regulating different target genes including WNT1, Cyclin G1, MDR, ADAM17, CUTL1, and AKT3 in HCC [130,131]. Associations of miR-122 expressed in liver and anti-oxidant genes, such as heme oxygenase 1 (HMOX-1), NAD(P)H, quinone oxidoreductase-1 (NQO1), and growth factor erv1-like (GFER1) in liver tissue specimens obtained from patients with chronic hepatitis B, have been uncovered. A significant positive association between expression of NQO1 and miR-122 was determined [35]. NQO1 is a multifunctional antioxidant enzyme and exceptionally versatile cytoprotective agent that regulates the proteasomal degradation of specific antioxidant proteins, such as nuclear factor erythroid 2-like 2 (Nrf2) [132], one of the major mediators of inflammation and a transcription factor. Nrf2 promotes the expression of antioxidant as well as cytoprotective genes, resulting in anti-inflammatory effects [133].

Perfluorooctanesulfonate (PFOS) has been widely used in commercial applications as a surfactant and stain repellent. PFOS has been shown to cause liver damage (including liver tumors) in experimental animals through interactions with peroxisome proliferator-activated receptor α (PPARα) and constitutive androstane receptor (CAR)/pregnane X receptor (PXR). Further studies have highlighted the ability of PFOS to disrupt thyroid function and induce thyroid hormone alterations, leading to hypothyroxinemia [78,134]. Assessment of changes in miRNA levels in rats with PFOS-induced hypothyroxinemia revealed that three members of the miR-200 family were the most significantly increased while miR-122 and miR-21 showed the greatest decrease in expression. Moreover, expression of the miR-23b/27b/24 cluster was decreased in PFOS-treated animals [78]. Consistently, experiments by our group demonstrated upregulation of miR-122, miR-21, and miR-24 by thyroid hormone treatment in HCC cells (Table 2).

Among the miRNAs affected by thyroid hormone, members of the miR-200 family were markedly enhanced in hepatic cells following hydrogen peroxide (H_2_O_2_) treatment. Among these, miR-200-3p modulates the H_2_O_2_-mediated oxidative stress response by targeting mitogen-activated protein kinase 14 (p38α). p38α acts as a stress-activated protein kinase that negatively regulates tumorigenesis by acting on cell apoptosis, survival, and stress response. p38α inhibition leads to increased ROS levels in liver cells through repression of Nrf2, a master regulator of antioxidant and detoxifying genes [40]. These results support a hepatoprotective role of thyroid hormone through effects on the pathway of oxidative stress-induced miR-200 to repress p38α and Nrf2.

MiR-92a is highly expressed and specifically altered in HBV/HCV-related HCC [135,136]. This miRNA plays a critical role in HCC proliferation and invasion and could serve as a novel therapeutic target via repression of Forkhead Box A2 (FOXA2) [137,138]. Clinical association analysis revealed a correlation of high expression of miR-92a with poor prognostic characteristics of HCC. Diagnostic efficacy of a combination of miR-92a and AFP was powerful for HCC, in particular screening of early tumor and low-level AFP patients [139]. A combination of the tumor suppressor gene phosphatase and tensin homolog (PTEN) and miR-92a also provided significant clinical value for early diagnosis and prognosis of HCC based on their significant negative correlation in HCC and para-cancerous tissue [140]. MiR-92a has been shown to promote tumor growth of HCC by targeting F-box and WD repeat domain-containing 7 (FBXW7) and may serve as a novel prognostic biomarker and therapeutic target [141]. ROS trigger DNA oxidation leading to multiple modifications in DNA bases, among which 8-OHdG is the most frequent [142]. 8-OHdG induces point mutations in DNA strands and accumulates in DNA to cause mispairing, resulting in mutagenic and potentially carcinogenic activity. HCC tissues are frequently characterized by increased oxidative damage, which contributes to acceleration of telomere shortening and telomerase activation in cancer cells. The telomere acts as a protective cap at the ends of chromosomes and telomere shortening promotes chromosomal instability [143,144]. Oncogenic miR-92a expression is significantly correlated with telomerase activity and 8-OHdG levels in HCC tissues, indicating a link with ROS-mediated oxidative DNA damage [32]. The pre-mRNA-splicing factor, SLU7, is essential for HCC cell viability. SLU7 expression is reduced in HCC cells, and its depletion triggers autophagy-related cellular apoptosis in association with generation of ROS. Low expression of SLU7 leads to altered splicing of the C13orf25 primary transcript and reduced expression of its miR-17-92a constituents, leading to upregulation of its target genes, CDKN1A (P21) and BCL2L11 (BIM), and mediators of pro-survival and tumorigenic activities [90]. Previous studies have shown that miR-92a and its cluster miR-17, miR-18a, and paralog, miR-20b, are downregulated by the thyroid hormone in HCC cells (Table 3). MiR-92 also plays a key regulatory role in neovascularization and is predicted to target Sirtuin-1 (Sirt1) [89], a NAD^+^-dependent deacetylase with potential anti-oxidative stress activity in vascular endothelial cells. The mechanisms underlying the protective effects involve Sirt1/FOXOs, Sirt1/NF-κB, Sirt1/NOX, Sirt1/SOD, and Sirt1/eNOS pathways [145]. Several other miRNAs, such as miR-181, miR-138, and miR-199, suppress Sirt1 in different cells/tissue types. Among these, miRNA-181 is upregulated under conditions of a high-fat diet and is reported to suppress Sirt1 and impair insulin sensitivity in liver [41,87]. Similar results have been reported for miR-200 and miR-199 in the DEN model [88]. Data from our qRT-PCR array disclosed downregulation of miR-181 and miR-199 by thyroid hormone (Table 3) [38]. Notably, miR-181 is inversely correlated with TRβ1 in human cirrhotic peritumoral tissue, compared to normal liver [146]. These findings support the theory that thyroid hormone decreases oxidative stress via repression of miR-181 and miR-199 to increase the target gene Sirt1 expression in liver.

MiR-206 is downregulated during tumorigenesis and plays an important role in modulating the growth of multiple HCC cells via targeting cyclin-dependent kinase 9 (CDK9), which stimulates the production of abundant prosurvival proteins, leading to impaired cancer cell apoptosis [147]. Overexpression of miR-206 has been shown to inhibit proliferation, invasion, and migration of the HCC cell lines HepG2 and Huh7. Conversely, inhibition of miR-206 inhibition enhances expression of protein tyrosine phosphatase 1B (PTP1B) that plays an oncogenic role in HCC, in HepG2, and Huh7 cells [148]. MiR-206 also directly targets the c-Met gene for silencing and restoration of c-Met expression reverses the inhibitory effect of miR-206 on HCC [149]. Nrf2, involved in cellular antioxidant defense systems, protects against excessive ROS damage to macromolecules and consequent senescence and apoptosis. Upregulation of Nrf2-dependent antioxidant and metabolic genes and significantly reduced miR-1 and miR-206 expression in lung tumors are associated with reduced survival in patients with lung adenocarcinoma [150]. MiR-206 is involved in thyroid hormone-mediated regulation of lipid metabolism in HepG2 cells, and its expression is suppressed in patients with hyperthyroidism, indicating a role in thyroid hormone-induced disorders of lipid metabolism in the liver [151].

### 2.5. Thyroid Hormone Promotes Oxidative Stress in Hepatocytes by Regulating microRNAs

Thyroid hormone not only regulates miRNAs to prevent oxidative stress in hepatocytes, but also exerts effects on miRNAs that result in increased oxidative stress-induced damage to liver. Below, we have discussed a few examples of miRNAs positively correlated with oxidative stress and regulated by thyroid hormone.

MiR-128 is downregulated in HCC and suppresses cell proliferation through inducing G1 phase cell arrest via regulating phosphoinositide 3 kinase regulatory subunit 1 (PIK3R1) expression, which inhibits the phosphatidylinositol 3-kinase (PI3K)/AKT signal pathway [152]. In addition, miR-128 significantly inhibits HCC cell metastasis and stem-cell like properties through direct targeting of integrin alpha 2 (ITGA2) and integrin alpha 5 (ITGA5) [153]. Parkinson disease protein 7 (PARK7/ DJ-1) expression is elevated in various tumors and related to the survival of tumor cells under adverse stimuli, including oxidative stress. DJ-1, also known as Parkinson’s disease-associated protein (PDAP), performs multiple functions, including cysteine protease, anti-oxidative stress reaction, and tumorigenesis activities [154,155]. MiR-128 is downregulated and negatively correlated with DJ-1, which is a direct target of the miRNA, in HCC cells [42]. Dox also markedly upregulates miR-128 and downregulates Sirt1 expression by direct targeting and affecting the expression of other antioxidant proteins, such as Nrf2, Keap1, Sirt3, NQO1, and HO-1, leading to excessive oxidative stress in liver [43]. In our qRT-PCR array, the tumor suppressor, miR-128, was upregulated by thyroid hormone, suggesting a correlation between thyroid hormone and miR-128-affected antioxidant genes, such as Sirt1 (Table 2).

According to our qRT-PCR data, thyroid hormone enhances miR-128 to suppress Sirt1 expression in liver. MiR-29a and miR-29c are also upregulated by the thyroid hormone and associated with Sirt1 expression (Table 2). Previous experiments have shown that miR-29 controls the hepatic lipogenic process through regulation of anti-lipogenic transcription factor aryl hydrocarbon receptor (AHR) and Sirt1 in liver [44]. MiR-29a suppresses cell proliferation through direct targeting of Sirt1 in HCC [82]. PU box binding protein (PU.1) is a critical transcription factor involved in many pathological processes. In PU.1-deficient mice, miR-34a and miR-29c are highly expressed and regulate Sirt1 expression in hepatic stellate cells to resistant hepatic fibrosis [79]. The data suggest that the thyroid hormone may suppresses the anti-oxidative stress reaction by miR-128, miR-29, miR-29a, and miR-29c to direct targeting Sirt1 and indirect effect other anti-oxidative stress genes expression.

In HCC cells and tissues, miR-21 is upregulated and positively associated with cell migration and invasion abilities. Krueppel-like factor 5 (KLF5) acts as a tumor inhibitor in some cancer types. In an earlier study, KLF5 expression was inhibited through direct targeting by miR-21, leading to the induction of migratory and invasive abilities in HCC [156]. Betulinic acid (BA) is a pentacyclic triterpene that possesses potential pro-apoptotic activities through increasing mitochondrial ROS generation. Mitochondrial dysfunction activates the molecular apoptotic events leading to cell death in HCC. BA suppresses superoxide dismutase 2 (Sod2) expression through upregulation of miR-21, leading to mitochondrial ROS accumulation and apoptosis in HCC [45]. MiR-21 is reported to be activated via thyroid hormone-receptor interactions at the native TRE site in the promoter region [75]. The thyroid hormone may thus have a similar function as BA in increasing mitochondrial ROS generation and mitochondrial dysfunction through miR-21 expression in HCC.

MiR-196 is readily released in body fluids and blood during HBV/HCV-associated hepatitis as well as metabolic, alcohol-associated, drug-induced, and autoimmune hepatitis. Liver-specific miR-196 is a potential indicator of liver injury (mainly apoptosis, necrosis, and necroptosis) or hepatitis, showing variable expression during acute/fulminant, chronic, liver fibrosis/cirrhosis, and HCC [157]. Bach1, a basic leucine-zipper mammalian transcriptional repressor, negatively regulates HMOX1, a key cytoprotective enzyme with antioxidant and anti-inflammatory activities. MiR-196 significantly downregulates Bach1, leading to upregulation of HMOX1 gene expression and inhibition of HCV expression, further affecting oxidative stress and liver injury induced by HCV [46]. Data from our qRT-PCR experiments suggest that miR-196 is a potential oncogenic miRNA downregulated by thyroid hormone (Table 3).

MiR-199 family members (miR-199a/b-5p) are downregulated in HCC. Notably, the lower expression of miR-199a is also associated with poorer overall survival of HCC patients. MiR-199a overexpression in HCC cell lines is reported to inhibit cell proliferation, migration, and invasion. The miR199a family suppresses Rho-associated coiled-coil kinase 1 (ROCK1) post-transcriptionally to inhibit PI3K/Akt signaling, which is necessary for HCC proliferation and metastasis [158]. Moreover, miR-199 targets and negatively regulates X-box binding protein 1 (XBP1) and affects cyclin D, which is associated with cell cycle regulation in HCC cells [159]. Bile salts retained within the liver play a major role in liver injury during cholestasis and trigger cellular stress events, including protein misfolding, DNA damage, endoplasmic reticulum (ER) stress, and oxidative stress, that may result in cell death and pathogenesis of several liver diseases [160]. Another study reported elevated miR-199a-5p levels in bile acid-stimulated cultured hepatocytes of liver from bile duct-ligated mice. Elevated miR-199-5p disrupted sustained ER stress and prevented hepatocytes from undergoing bile acid-induced cell death, supporting the potential of this miRNA as a target for clinical approaches aiming to protect against liver toxicity from bile salts in hepatocytes [33]. Analysis of the association between thyroid hormone and miR-199 unexpectedly revealed a negative correlation between the two molecules. MiR-199/miR-214 are clustered and located on opposite strands of the Dynamin3 gene (DNM3). Under most conditions, while clusters have the same performance, but the thyroid hormone exerts differential effects on the two molecules, which seems to be due to the existence between miR-199 and miR-214 additional positivity TRE affecting miR-214 [38].

## 3. Discussion

The specific roles of thyroid hormone in different human cancer types are controversial. A number of investigators have reported that thyroid hormone promotes development of various cancers, whereas others suggest a tumor suppressor role [161,162,163,164]. Accumulating evidence from animal models and epidemiologic studies indicate an association between higher thyroid hormone levels and prevention of liver diseases, supporting the suppressor role of thyroid hormone and its receptor in HCC [25,38,164,165,166]. Moreover, clinical findings support a positive correlation between hypothyroidism and HCC development [167,168,169]. Oxidative stress-induced liver inflammation is the most important factor for HCC progression [1,170]. Oxidative stress is also related to thyroid hormone derangement, with the hormone reported to influence the antioxidant level or generation of ROS. Hyperthyroidism and hypothyroidism have been shown to be associated with oxidative stress in acute and chronic nonthyroidal illness syndrome (NTIS) [171,172].

In this report, we have discussed the involvement of a range of miRNAs in correlation with thyroid hormone and oxidative stress in HCC. Inconsistent results have been obtained from multiple studies on the role of the thyroid hormone in multiple cancer types. Based on the collective findings, thyroid hormone clearly regulates the expression of different miRNAs either directly or indirectly to affect oxidative stress (HCV/HBV-induced or DEN, Dox-induced) in liver. As shown in Figure 1, thyroid hormones can influence oxidative stress-induced hepatocarcinogenesis mediated by miRNAs, specifically, via upregulating miR-214, miR-122, and miR-206 in HCC. Several other studies indicate these miRNAs act as tumor suppressors in HCC. Simultaneously, these miRNAs regulate different oxidative stress-related genes that participate in liver cell antioxidant capacity, including Bim, NQO1, and Nrf2. A number of oncogenic miRNAs have been shown to be downregulated by thyroid hormone in HCC, including miR-200, miR-92a, and miR-181. Their target genes, p38α, 8-OHdG, and Sirt1, participate in oxidative stress. Multiple studies have shown that thyroid hormones participate in the regulation of miRNA expression to prevent excessive generation of ROS (miR-122, miR-200, miR-206) and reduce DNA damage (miR-214, miR-92a, miR-181) in hepatocytes.

Conversely, an oncogenic role of thyroid hormone in HCC has been reported by other investigators [14,173]. The thyroid hormone is reported to induce oxidative stress through enhancing the expression of different miRNAs in liver, such as miR-128, miR-29a/c, and miR-21, which directly regulate anti-oxidant genes, such as DJ-1, Sirt1, and Bach1, or affect related antioxidant genes, including Nrf2, Keap1, Sirt3, NQO1, and HO-1.

The role of thyroid hormone in various human organs is complex, especially under conditions of oxidative stress. In patients with hypothyroidism, high plasma levels of NO and malondialdehyde (MDA), a marker of oxidative stress, were measured in hepatic vein, along with lower activity of paraoxonase-1 (PON-1), a liver enzyme with antioxidant features. Data from this study further indicated that increased oxidative stress in hypothyroidism is primarily attributed not only to a decrease in antioxidant levels, but also effects on lipid metabolism [174,175]. The collective evidence suggests that thyroid hormone has potential antioxidant activity, at least in the liver [7,73]. The same findings were reported in experimental animals. Thyroid hormone signaling was altered following stimulation with various stress signals and played a crucial role in response to stress, post-stress recovery, and tissue repair by reducing the inflammatory response associated with NF-κB and STAT3 activation as well as the acute phase response [176,177,178]. Hepatic autophagy regulates lipid metabolism through elimination of triglyceride accumulation in liver, prevents the development of steatosis, and reduces oxidative stress [179]. Mitochondrial balance is a precisely regulated process that influences cellular homeostasis. Since activation of hepatic mitophagy eliminates the lipid content and oxidative stress, its dysregulation is implicated in progression of NAFLD. Dysregulation of autophagy and defective mitochondrial homeostasis contribute to hepatocyte injury and liver-related diseases [180]. In a murine model, disruption of thyroid hormone production led to a marked increase in progression of DEN-induced HCC. DEN triggers generation of ROS, resulting in oxidative stress and cellular injury, and consequently, progression of HCC. Data from this model indicate that thyroid hormone promotes autophagy via induction of the DAPK2-SQSTM1 cascade, thus protecting hepatocytes from DEN-induced hepatotoxicity or carcinogenesis. Furthermore, thyroid hormone is reported to participate in regulation of lipid metabolism through a chromosome 19 open reading frame 80 (C19orf80)-activated autophagic process in HCC [99,101,181].

Although the role of thyroid hormone in relation to oxidative stress in HCC remains a controversial issue, abundant reports support antioxidant over pro-oxidant activity, which prevents induction of liver cancer progression by oxidative stress in hepatocytes. This situation is similar to that of the antioxidant genes Sirt1 and Nrf2. Sirt1 and Nrf2 are oppositely regulated by thyroid hormone in association with different miRNAs. However, the majority of reports indicate that the thyroid hormone activates Sirt1 [182,183]. Moreover, the thyroid hormone positively affects FOXO1 in addition to Sirt1 to stimulate genes that enhance the autophagic process [184,185,186,187]. Several animal studies have provided evidence to support a positive association of thyroid hormone with Nrf2, a key redox-sensitive transcription factor of antioxidant genes, in the liver. Thyroid hormone not only upregulates Nrf2, but also promotes antioxidant gene expression, since Nrf2 translocates from the cytosol to nucleus, mediating hepatic cytoprotection [188,189,190,191].

In this review, we have discussed several miRNAs associated with oxidative stress in HCC. Importantly, the relationships among miRNAs, thyroid hormone, and oxidative stress have been comprehensively explored. Despite conflicting results in the literature, thyroid hormone is considered a protective factor overall in hepatocytes. Thyroid hormone may aid in maintaining the normal environment of hepatocytes through effects on lipid metabolism and mitochondrial activity. Moreover, thyroid hormone protects against liver injury by reducing oxidative stress induced by harmful chemicals or HBV/HCV. Further studies should focus on the development of thyroid hormone analogs beneficial for human health.

## Figures and Tables

**Figure 1 ijms-20-05220-f001:**
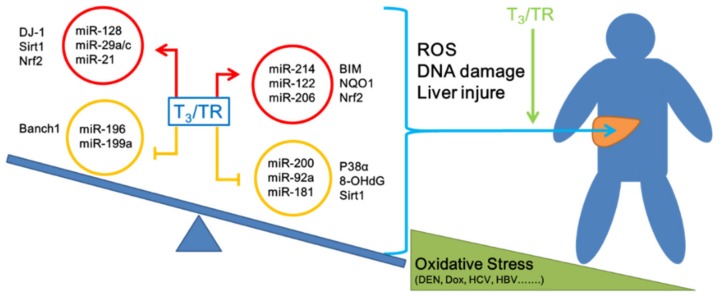
Thyroid hormones affect oxidative stress-induced hepatocarcinogenesis through effects on miRNAs. Oxidative stress is a risk factor associated with liver cancer. Among them, miRNAs strongly related to physiological significance are also involved. We analyzed the associations of thyroid hormones with oxidative stress and miRNAs in liver cells. While miRNAs related to promotion of resistance to oxidative stress were also affected by thyroid hormones, based on empirical evidence from other experimental animal and clinical studies, we believe that thyroid hormone plays a largely hepatoprotective role under conditions of oxidative stress. (The red arrow is a positive association, the yellow T bar is a negative association, blue arrow indicates the miRNAs affected liver genes, and green arrow is thyroid hormone associated-miRNAs affected liver genes)

**Table 1 ijms-20-05220-t001:** Oxidation stress-related microRNAs.

microRNA	Correlative with Oxidative Stress	Ref.
miR-34a-5p miR-1915-3p miR-638 miR-150-3p	Associated with oxidative stress-related apoptosis	[30]
miR-92	Correlated positively with telomerase activity, 8-OHdG Target to anti-oxidative gene Sirt1	[32]
miR-199a/b	Prevents the liver cell oxidative stress induced by bile acid Target to Sirt1	[33]
miR-122	Correlative with HCV/ HBV infection Positive association with antioxidant enzyme NQO1	[34] [35]
miR-26a	Affecting liver lipid metabolism	[36]
miR-155	Affecting liver lipid metabolism	[37]
miR-214	Associated with oxidative stress-related apoptosis Target to ATF4 and EZH2	[38] [39]
miR-200	Target to p38α and repression anti-oxidative gene Nrf2	[40]
miR-181	Target to Sirt1 and impair insulin sensitivity	[41]
miR-128	Target to DJ-1 Target to Sirt1	[42] [43]
miR-29a/c	Controls the hepatic lipogenic process	[44]
miR-21	Leading to mitochondrial ROS accumulation	[45]
miR-196	Downregulates Bach1, and inhibition of HCV expression	[46]

* Potential functions of miRNAs related to oxidative stress.

**Table 2 ijms-20-05220-t002:** MicroRNAs positively associated with thyroid hormones in HepG2 liver cancer cell lines.

	miRNAs Positively Affected by Thyroid Hormones *							
	HepG2-TRα1				HepG2-TRβ1			
TH/microRNAs	MicroRNA	Fold	HCC/ROS Ref.	TH	MicroRNA	Fold	HCC/ROS Ref.	TH
Three times repetitive experiments	miR-122	10.54	[35,77]	[78]	miR-29c	4.21	[44,79]	
	miR-152	3.42			miR-214	3.50	[38,80,81]	[38]
	miR-139-5p	10.38			miR-202	2.41		
	miR-128a	71.90	[43]					
	miR-139-3p	3.85						
	miR-548d-3p	3.09						
	miR-140-3p	2.77						
Two times repetitive experiments	miR-143	6.20			miR-193b	2.99		
	miR-210	5.22			miR-139-5p	3.12		
	miR-365	5.53			miR-210	2.52		
	miR-135b	4.38			miR-323-3p	4.12		
	miR-148a	5.16			miR-22	2.54		
	miR-193b	3.30			miR-29a	2.18	[44,82]	
	miR-125a-3p	2.92			miR-29b-1 *	3.30		
	miR-29a	3.15	[44,82]		miR-193a-3p	3.34		
	miR-24	2.40	[83]	[78]	miR-139-3p	2.22		
	miR-372	3.57			miR-510	2.32		[75,78]
	miR-372	5.07			miR-21 *	2.22	[45,84,85,86]	
	miR-188-3p	3.05						
	miR-100	4.11						
	miR-126	2.35						
	miR-21	3.30	[45,84,85,86]	[75,78]				

* HepG2 hepatoma cell lines overexpressing TRα1 or TRβ1 were treated with thyroid hormone (T3; 20 nM). After 24 h, qRT-PCR array analysis of microRNA (miRNA) expression was performed. The specified miRNAs were positively affected (>2-fold) upon thyroid hormone stimulation and selected candidates were identified from at least two times repetitive experiments. The references are to indicate oxidative stress (HCC/ROS) or thyroid hormone (TH) related miRNAs in liver cancer.

**Table 3 ijms-20-05220-t003:** MicroRNAs negativity associated with thyroid hormones in HepG2 liver cancer cell lines.

	miRNAs Negativity Affected by Thyroid Hormones *							
	HepG2-TRα1				HepG2-TRβ1			
TH/microRNAs	MicroRNA	Fold	HCC/ROS Ref.	TH	MicroRNA	Fold	HCC/ROS Ref.	TH
Three times repetitive experiments	miR-184	0.22			miR-455-3p	0.22		
	miR-455-3p	0.12			miR-148a	0.36		
	miR-499-3p	0.20			miR-425 *	0.24		
	miR-221	0.33			miR-187	0.27		
	miR-181b	0.30	[87]		miR-429	0.41		
	miR-130b	0.34		[76]				
	miR-149	0.35						
	miR-17	0.34	[85]	[74]				
Two times repetitive experiments	miR-425 *	0.22			miR-106a	0.23		
	miR-20a	0.31			miR-199a-5p	0.22	[33,87,88]	[38]
	miR-377	0.42			miR-548d-5p	0.24		
	miR-15b	0.43			miR-146a	0.31		
	miR-516a-5p	0.29			miR-221	0.27		
	miR-652	0.49			miR-30a *	0.35		
	miR-550	0.26			miR-499-3p	0.32		
	miR-18a	0.23			miR-888	0.27		
	miR-106a	0.28			miR-100	0.33		
	miR-628-3p	0.34			miR-339-3p	0.45		
	miR-146a	0.36			miR-18a	0.39		
	miR-181c	0.41	[87]		miR-18b	0.24		
	miR-92a	0.36	[32,89,90]		miR-10a	0.25		
	miR-106b	0.38			miR-421	0.30		
	miR-487b	0.35			miR-525-3p	0.41	[74,85]	
	miR-570	0.40			miR-17	0.37	[85,90]	[74]
	let-7d	0.44			miR-542-5p	0.33	[46,85]	
	miR-15b *	0.44			miR-196a *	0.42	[46]	
					miR-196b	0.46		
					miR-19a	0.46	[87]	
					miR-181d	0.32		
					miR-20b	0.40		

* HepG2 hepatoma cell lines overexpressing TRα1 or TRβ1 were treated with thyroid hormone (T3; 20 nM). After 24 h, qRT-PCR array analysis of microRNA (miRNA) expression was performed. The specified miRNAs were negatively affected (<0.5-fold) upon thyroid hormone stimulation and selected candidates were identified from at least two times repetitive experiments. The references are to indicate oxidative stress (HCC/ROS) or thyroid hormone (TH)-related miRNAs in liver cancer.

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
