# Peer review of "Roles of Thyroid Hormone-Associated microRNAs Affecting Oxidative Stress in Human Hepatocellular Carcinoma"

_ijms, 2019, doi:10.3390/ijms20205220_

Round 1

Reviewer 1 Report

The manuscript is well written and informative. However, it would benefit from a more systematic presentation of the data in tabular form. For example, data described in the subheading 2.2. (line 111 through 170) should also be presented in a table showing the supporting evidence for miRs associated with oxidative stress in HCC.

For Tables 1 and 2, the titles should be reformulated in order to fully relay the content. For example: “MicroRNAs positively associated with thyroid hormones in HepG2 liver cancer cell lines”.

The format of these tables also needs to be modified in order to be more readable. The rows “Three times” and “Two times” should be fully spelled out because there is enough space.

I appears that this data was already presented in refs 50, 59-61 from the same group. Make sure it is not presented exactly in the same way, or just duplication of the same data. If this is a systematic presentation of the entire data gathered from all those references, then state so.

Author Response

Response to Reviewer 1 Comments:

1-1

The manuscript is well written and informative. However, it would benefit from a more systematic presentation of the data in tabular form. For example, data described in the subheading 2.2. (line 111 through 170) should also be presented in a table showing the supporting evidence for miRs associated with oxidative stress in HCC.

Authors’ response:

We have integrated the functions of miRNAs associated with oxidative stress into a table for readers easier to understand. (p.3, Table.1)

1-2

For Tables 1 and 2, the titles should be reformulated in order to fully relay the content. For example: “MicroRNAs positively associated with thyroid hormones in HepG2 liver cancer cell lines”.

Authors’ response:

We have modified the title of the Table2 & 3 (revised version) as suggested. (p.6, line.240; p.7, line.247)

1-3

The format of these tables also needs to be modified in order to be more readable. The rows “Three times” and “Two times” should be fully spelled out because there is enough space.

Authors’ response:

We have modified the format of Tables for readers easier to understand. (p.6, Table.2, 19; p.7, Table.3)

1-4

I appears that this data was already presented in refs 50, 59-61 from the same group. Make sure it is not presented exactly in the same way, or just duplication of the same data. If this is a systematic presentation of the entire data gathered from all those references, then state so.

Authors’ response:

We have modified the sentences to describe our previous data to support the TR or microRNAs associated oxidative stress (p.6, line.236-237)

Reviewer 2 Report

The review "roles of thyroid hormone-associated microRNAs affecting oxidative stress in human hepatocellular carcinoma” is a review of thyroid hormone effects on those miRNAs that show an important role in oxidative stress and HCC. In general the review is interesting and well organized but I have some concerns:

Major point:

- This review explains how thyroid hormones affects microRNAs implicated in oxidative stress and HCC. However, authors almost do not explain thyroid hormones and its mechanisms of action. This review could be read by authors that works with HCC, miRNAs and/or oxidative stress and without a solid background in thyroid hormone and thyroid hormone receptors actions. A more detailed explanation of thyroid hormone and its actions (and mechanisms of action) should be added.

Minor points:

- Authors explain in detail thyroid hormone effects on apoptosis (paragraph 6 of section 2.3). In the next paragraph they explain the posible role of thyroid hormones in apoptosis and miR-214. In my opinion authors focus too much attention to thyroid hormone and apoptosis and less attention to miRNAs. They should give more information about miRNAs and HCC and apoptosis or try to reduce the information of the paragraph 6 of section 2.3.

- In line 254 authors say “The tumor suppressor activity ofmiR-214” and they should add a space between of and miR-214.

In conclusion, from my point of view this review is interesting but these points should be clarified for publication in IJMS.

Author Response

Response to Reviewer 2 Comments:

2-1

This review explains how thyroid hormones affects microRNAs implicated in oxidative stress and HCC. However, authors almost do not explain thyroid hormones and its mechanisms of action. This review could be read by authors that works with HCC, miRNAs and/or oxidative stress and without a solid background in thyroid hormone and thyroid hormone receptors actions. A more detailed explanation of thyroid hormone and its actions (and mechanisms of action) should be added.

Authors’ response:

We have added more information about the role of thyroid hormone receptors action according to the reviewer’s suggestion. (p.5, line.177-206; p.6, line.207-220)

2-2

Authors explain in detail thyroid hormone effects on apoptosis (paragraph 6 of section 2.3). In the next paragraph they explain the posible role of thyroid hormones in apoptosis and miR-214. In my opinion authors focus too much attention to thyroid hormone and apoptosis and less attention to miRNAs. They should give more information about miRNAs and HCC and apoptosis or try to reduce the information of the paragraph 6 of section 2.3.

Authors’ response:

We have discussed more about the miRNAs associated with the apoptotic pathway in liver cancer as suggested. (p.9, line.299-310)

In line 254 authors say “The tumor suppressor activity ofmiR-214” and they should add a space between of and miR-214.

Authors’ response:

We have fixed the error. (p.9, line.314-315)